# Effect of Potato Vine and Leaf Mixed Silage to Whole Corn Crops on Rumen Fermentation and the Microbe of Fatten Angus Bulls

Siyu Zhang [1,†], Jiajie Deng [1,†], Yafang Cui [1], Lina Wang [1], Yingqi Li [1], Xianli Wang [1], Shengnan Min [1], Huili Wang [1], Qianzi Zhang [1], Peiqi Li [1], Yawen Luo [1], Xinjun Qiu [2], Yang He [1], Binghai Cao [1] and Huawei Su [1,*]

1   State Key Laboratory of Animal Nutrition, College of Animal Science and Technology, China Agricultural University, Beijing 100107, China; s20213040648@cau.edu.cn (S.Z.); 18811560713@163.com (J.D.); s20203040609@cau.edu.cn (Y.C.); linalinaw@163.com (L.W.); liyingqi1230@163.com (Y.L.); w18832607007@163.com (X.W.); sy20223040817@cau.edu.cn (S.M.); wanghlzn@126.com (H.W.); dili1899@163.com (Q.Z.); peppalily@163.com (P.L.); luoyawen230023@163.com (Y.L.); heycau@163.com (Y.H.); caobh@cau.edu.cn (B.C.)
2   College of Animal Science and Technology, Hainan University, Haikou 570228, China; qiuxinjun@hainanu.edu.cn
*   Correspondence: suhuawei@cau.edu.cn
†   These authors contributed equally to this work.

**Abstract:** The objective of this study was to evaluate the effects of potato vine and leaf mixed silage (PVS) on rumen fermentation and the microbe in ruminants and to improve the utilization of PVS resources in ruminants through in vitro gas production and feeding trials. The experiment was divided into three groups: PVS1 (50% corn + 50% potato vine and leaf silage), PVS2 (75% potato vine and leaf + 15% rice straw + 10% cornmeal silage) and whole-plant corn silage (CS). The in vitro gas production results showed that there was a significant reduction in PVS groups in the indexes of total gas ($p < 0.05$) and $CH_4$ production ($p < 0.05$). The digestibility of dry matter ($p < 0.05$), neutral detergent fiber (NDF) ($p < 0.05$), and acid detergent fiber (ADF) ($p < 0.05$) at 48 h were decreased in the PVS group. For the rumen fermentation indexes, the pH ($p < 0.05$), microbial crude protein (MCP) ($p < 0.05$), and acetate to propionate ($p < 0.05$) showed an increase in the PVS group, but a decrease in the total volatile fatty acid concentration ($p < 0.05$). In the feeding trial, different silages in diets had no significant effect on the rumen fermentation indexes ($p > 0.05$). In the rumen microbe composition, the PVS diet significantly reduced the abundance of Prevotella ($p < 0.05$) compared with the CS diet group. The PVS2 diet significantly increased the abundance of the *Lachnospiraceae_XPB1014_group* ($p < 0.05$) and *Bacteroidales_bacterium_Bact_22* ($p < 0.05$) compared with the CS diet group. In conclusion, PVS had no negative effect on rumen fermentation characteristics and rumen microbial flora and could significantly reduce rumen gas production compared to CS, positively mitigating animal husbandry $CH_4$ emissions and environmental protection.

**Keywords:** potato vine and leaf mixed silage; in vitro gas production; rumen fermentation; rumen microbe

## 1. Introduction

As one of the world's four major food crops, the potato (*Solanum tuberosum* L.) has been widely cultivated in more than 150 countries and regions. According to data released by the Food and Agriculture Organization (FAO), global potato production reached 376 million tons in 2021 [1]. In the potato production process, only potatoes are harvested. Potato production waste by-products after potato harvest include its vine and leaf with a high crude protein (CP) content (11–26% dry matter (DM) basis) [2–4]. However, raw potato vine and leaf can cause neurological and digestive dysfunction in mammals because it contains solanine (the main components are α-solanine and α-theophylline) [5,6]. The

withered yellow vine and leaf after potato harvest is then abandoned in farmland or burned, producing a lot of greenhouse gases. Therefore, how to utilize this agricultural by-product has become an urgent problem to solve the agricultural by-product sustainability requirement.

Studies have found that microorganisms can degrade α-solanine and α-chakonin into glycans during ensiling or rumen fermentation [4,7,8]. Therefore, the process of the potato vine and leaf mixed silage ensiling can detoxify α-solanine and α-chakonin. However, due to the low DM content and water-soluble carbohydrate (WSC) content [4], the potato vine and leaf are not suitable for ensiling [2,9]. Previous studies found that adding rice bran and wheat bran to the potato vine and leaf improved silage quality and had no adverse effects on rumen fermentation [10]. To facilitate the mix silage quality, adding a fermentation promoter can meet the requirement of WSC for ensiling [4,11]. Therefore, mixing the potato vine and leaf with whole-plant corn or adding corn starch is a good way to facilitate the fermentation process and improve the quality of mixed silage.

According to International Energy Agency (IEA) research, global methane emissions in 2022 reached 3,558,013 million tons [12]. The largest source of emissions included agricultural activities, accounting for 39.90% [12]. Additionally, the GHG effect capacity of $CH_4$ was 28 times that of $CO_2$ [13]. At the same time, in vitro, rumen gas production experiments found that the $CH_4$ production of potato vine and leaves was significantly lower than that of whole plant corn silage [14]. Therefore, replacing traditional silage with potato vine leaf mixing silage is a solution to reducing greenhouse gas emissions. This silage production reduces the waste of agricultural by-product resources and contributes to $CH_4$ emission reduction and sustainable development in animal husbandry, which has a positive impact on environmental protection.

Nearly all PVS research has mainly focused on silage nutrients and fermentation quality. There are few studies regarding the utilization of the potato vine and leaf mixed silage to replace whole plant corn silage in the beef cattle diet. Based on previous results, we hypothesized that the PVS substitution for CS might affect the rumen's digestibility, microbial composition, and the ruminant methane emissions of the Angus bull. This study aimed to explore the effect of potato vine and leaf mixed silage on rumen fermentation characteristics of Angus bulls through rumen fermentation in vitro and in vivo and provides a reference for measuring the effect and value of potato vine and leaf as a corn silage replacement and reducing greenhouse gas emissions.

## 2. Materials and Methods

### 2.1. Ensiling of Three Silages

The harvest and ensiling process of these three silages were conducted at Benwang Farm (106°028′ E 38°202′ N, Ningxia, China): a collaborative farm with the China Agricultural University. The whole-plant corn and potato vine and leaf were harvested and then processed at Benwang Farm from 5–15 September 2019 simultaneously.

The whole plant corn was harvested at the half to two-thirds milk line stage and was chopped with a silage harvester to a consistent length of 19 mm, sealed with an oxygen barrier film (Shanghai Comiy BioTechnology LTC, Shanghai, China) and stored in a silage pit with a density near 700 kg/m³ (210 kg DM/m³). Potato vine and leaf were harvested when most of the potato vine and leaf turned from green to yellow and withered, using a potato harvester to separate the potato fruit from the potato vine and leaf, collecting the vine and leaf by a 4-wheeled potato harvester (Meinuo 1120A, MeinuoZhongji Meinuo Machinery & Equipment Co., Langfang City, China) and wrapping silages with a radius of 40 cm, a height of 80 cm and a density of nearly 800 kg/m³ (PVS1: 205 kg DM/m³, PVS2: 285 kg DM/m³). After ensiling, we checked weekly for breaks and leaks.

Finally, three treatments were obtained: (1) whole-plant corn silage (CS); (2) potato vine and leaf mixed silage 1 (PVS1): 50% potato vine and leaf + 50% whole corn; (3) potato vine and leaf mixed silage 2 (PVS2): 75% potato vine and leaf + 15% rice straw + 10% corn meal. The three silages were preserved following the same silage management methods.

The silage insulation film opened after 60 d of continuous fermentation, and the wrapped silage was sampled on the top, middle and bottom layers to analyze the nutrition values. The nutritional values of the three silages are shown in Table 1.

**Table 1.** Chemical composition of three different silages.

| Item [1] | CS | PVS1 | PVS2 |
|---|---|---|---|
| pH-value | 4.48 | 4.69 | 4.87 |
| Nutritive values, % DM [2] | | | |
| Dry matter | 30.07 | 25.59 | 35.71 |
| CP | 6.91 | 7.84 | 8.45 |
| EE | 2.46 | 3.30 | 3.09 |
| NDF | 44.04 | 43.42 | 41.79 |
| ADF | 24.45 | 30.00 | 27.48 |
| Ash | 4.94 | 10.42 | 18.48 |
| Starch | 31.81 | 19.95 | 24.15 |
| Metabolic energy, MJ/kg [3] | 11.08 | 10.51 | 9.38 |

[1] CS: Whole-plant corn silage; PVS1: Potato vine and leaf mixed silage 1; PVS2: Potato vine and leaf mixed silage; [2] CP: Crude protein; EE: Ether extract; NDF: Neutral detergent fiber; ADF: Acid detergent fiber; [3] ME was based on the total digestible nutrients of each feed ingredient. (ME = $0.82 \times 4.409 \times$ TDN, TDN (%) = $0.98 \times (100 - \text{NDFn-CP} - \text{Ash} - \text{EE} + \text{IADICP}) + \text{kdCP} \times \text{CP} + 2.25 \times (\text{EE} - 1) + 0.75 \times (\text{NDFn} - \text{ADL}) \times [1 - (\text{ADL(NDFn)}0.667] - 7)$, referring to NRBC (2016) [15].

### 2.2. In Vitro Incubation and Degradability Measurement

Three Angus gelding bulls ($844.37 \pm 17.2$ kg, Mean $\pm$ SEM) with permanent ruminal fistulas were selected as experimental donor animals for ruminal fluid (Fangshan Beef Cattle Experimental Base of China Agricultural University, Beijing, China). The diet formulation refers to the eighth edition of the Nutrient Requirements of Beef Cattle (2016) [15]. The diet composition and nutrition values are shown in Table 2. They were fed twice daily (07:00 and 16:00, GMT+8), with free drinking water and pre-feeding for 7 days. Feed samples were air-dried and crushed on a 1 mm sieve as a fermentation substrate. The rumen fluid of each cattle was collected 1 h before feeding in the morning through four layers of medical cheesecloth and was mixed in equal proportions. The mixed rumen fluid was transferred to a preheated 39 °C incubator (Shanghai Yiheng, Shanghai, China) for subsequent batch culture as an inoculant.

**Table 2.** The composition of the fistula cattle diet.

| Item% | Content% |
|---|---|
| Ground corn | 15.0 |
| Dried distiller's grains with soluble (DDGS) | 3.6 |
| Soybean meal | 4.2 |
| Jujube powder | 4.5 |
| Stone powder | 0.3 |
| $NaHCO_3$ | 0.6 |
| NaCl | 0.3 |
| $CaHPO_3 \cdot 2H_2O$ | 0.3 |
| 5% Premix [1] | 1.2 |
| Wheat straw | 70.0 |

[1] The premix contains 1 g of copper, 12 g of iron, 11 g of zinc, 1 g of manganese, 30 mg of selenium, 30 mg of iodine, 20 mg of cobalt, 450,000 IU of vitamin A, 13,000–100,000 IU of vitamin D3, and 2000 mg of vitamin E.

The in vitro disappearance and kinetic gas production rates of DM (IVDMD), NDF (IVNDFD), and ADF (IVADFD) were determined by the Pang method [16]. A total of 8 replicates were performed for 24 and 48 h per sample to detect $\text{IVDMD}_{24}$, $\text{IVDMD}_{48}$, $\text{IVNDFD}_{24}$, $\text{IVNDFD}_{48}$, $\text{IVADFD}_{24}$, and $\text{IVADFD}_{48}$. A total of 1 g of the substrate (DM) was weighed in a culture bottle. In total, 50 mL of a 39 °C pre-warmed buffer solution, as described in Supplementary Table S1 (pH 6.85), and 25 mL of filtered rumen fluid were

mixed into each culture bottle (glass bottle with chitin stopper, total volume 125 mL). Each bottle was then washed with $N_2$ to remove air and was immediately sealed with a chitin stopper and screw cap, connecting each bottle to AGRS-III (AGRS-III, China Agricultural University, Beijing, China) [17]. Each bottle was removed from the culture system after incubation at 39 °C for 24 h and 48 h, and the pH was determined using a portable pH meter (testo 205, Lenzkirch, Germany). To detect IVDMD, the biomass material was filtered in each bottle using a nylon bag drying at 65 °C for 48 h. Residues were used for further NDF and ADF analysis (ANKOM A2000i, NY Macedon, USA). The filtered digestive liquid of each bottle was collected for the detection of NH3-N, the microbial crude protein (MCP), and volatile fatty acids (VFA). The concentrations of MCP and $NH_3$-N were determined by Makkar and Verdouw methods [18,19]. The concentration of volatile fatty acids (VFA) in the supernatant was determined by gas chromatography (GC-2014 Shimadzu Corporation, Kyoto, Japan).

## 2.3. Animal Pens and Diets

Thirty-six 13-month-old Angus bulls were divided into 9 pens according to their weight of 403.22 ± 6.58 kg (Mean ± SEM). The three groups were fed to ensure that the DM intake was the same, and the proportions were adjusted according to the different DM contents of the three silages. As a result, the DM of the diet was 57.2% for CS, 54.4% for PVS1, and 62.5% for PVS2. The diet composition and nutrition values are shown in Table 3.

**Table 3.** Composition, proportions, and nutritional values of three different dietary silage groups.

| Item% | CS | PVS 1 | PVS 2 |
|---|---|---|---|
| Flaked corn | 21.85 | 21.59 | 22.08 |
| Soybean meal | 5.59 | 5.36 | 5.79 |
| Wheat bran | 3.73 | 3.57 | 3.86 |
| Cotton meal | 2.98 | 2.86 | 3.09 |
| 5% Premix [1] | 2.98 | 2.86 | 3.09 |
| Dried distiller's grains with soluble (DDGS) | 2.98 | 2.86 | 3.09 |
| Rapeseed meal | 2.01 | 1.93 | 2.08 |
| Probiotics [2] | 1.34 | 1.29 | 1.39 |
| Fermented feed [3] | 4.97 | 4.76 | 5.14 |
| Straw | 2.48 | 2.38 | 2.57 |
| Rice stalks | 6.21 | 5.95 | 6.43 |
| Silage | 42.89 | 45.25 | 40.83 |
| Nutritive values, % DM [2] | | | |
| Dry matter | 57.2 | 54.4 | 62.5 |
| Crude protein | 11.24 | 13.17 | 13.6 |
| Ether extract | 7.87 | 5.75 | 5.53 |
| Neutral detergent fiber | 28.35 | 28.76 | 27.7 |
| Acid detergent fiber | 17.00 | 17.00 | 15.41 |
| Ash | 8.51 | 9.11 | 10.81 |
| Ca [4], % | 0.41 | 0.39 | 0.42 |
| P [4], % | 0.15 | 0.14 | 0.15 |
| Starch | 31.81 | 19.95 | 24.57 |
| Metabolic energy, MJ/kg [5] | 12.17 | 11.83 | 11.37 |

[1] 5% Premix composition per kg Vitamin A 120,000–200,000lU, Vitamin E > 550lU, D-Biotin 20.3 mg, Copper 0.16–0.5 g, Manganese 0.6–2.4 g, Selenium 1.6–10 mg, calcium 10.0–20.0%, vitamin D3 15,000–60,000lU, nicotinamide 2350 mg, iron 0.8–8.4 g zinc 1.5–3.0 g, iodine 4–20 mg, sodium chloride 10.0–20.0%, total phosphorus z 2.0%. [2] Probiotics are Stirling S-7001 (Guangdong VTR Bio-tech CO, LTD, Zhuhai, China). [3] Fermented feed is yeast culture (Jiangsu Yiyuantai Bio-tech CO., LTD, Suqian, China). [4] Calcium and phosphorus are estimates, not actual analysis values (Database of FeedComposition and Nutritive Values in China, Version 31 2020 China). [5] ME was based on the total digestible nutrients of each feed ingredient. (ME = 0.82 × 4.409 × TDN, TDN (%) = 0.98 × (100 − NDFn − CP − Ash − EE + IADICP) + kdCP × CP + 2.25 × (EE − 1) + 0.75 × (NDFn − ADL) × [1 − (ADL(NDFn)0.667] − 7), referring to NRBC (2016) [15].

*2.4. Sample Collection and Analysis*

From 87 to 88 days after the start of the feeding trial, the rumen fluid was extracted with a rumen catheter (Type-K0021, ANSCITECH Ltd., Wuhan, China). The rumen fluid of each cattle was collected with a syringe 1 h before feeding in the morning. To avoid saliva contamination, the first 100 mL of the rumen fluid was discarded and then packed into two 50 mL centrifuge tubes using a portable pH meter (testo 205, Lenzkirch, Germany) and was stored in liquid nitrogen for DNA extraction, and the later analysis of $NH_3$-N and VFA.

*2.5. DNA Extraction and 16s rRNA Gene Sequencing*

In this study, DNA extraction and amplicon were sequenced and referred to Yongjuan He's method [20]. Bioinformatics analysis was carried out with the assistance of BMK Cloud (Biomarker Technologies Co., Ltd., Beijing, China). The original data source for Illumina sequencing has been uploaded to NCBI with the serial number PRJNA958460.

*2.6. Gas Production Kinetics*

The cumulative gas production data were calculated, and the equation used was as below [21]:

$$GPt = \frac{A}{1 + \left(\frac{C}{t}\right)^B}, \tag{1}$$

$GP_{48}$ is the cumulative gas production (mL/g DM) at incubation time 48 (h).

$CH_4$ production was predicted according to VFA stoichiometry equations, and the equation used is shown below [22,23]:

$$\text{Predicted } CH_4 \text{ (mL)} = 22.4 \text{ (mL/mmol gas)} \times (0.5 \times \text{Acetate} - 0.25 \times \text{Propionate} + 0.50 \times \text{Butyrate} - 0.25 \times \text{Valerate}) \tag{2}$$

The average gas production rate (AGPR, mL/h) was calculated with A, B, and C, and the equation used is shown below [17]:

$$AGPR = A \times B / (4 \times C). \tag{3}$$

*2.7. Statistical Analysis*

All data were initially processed in Excel 2016 (Microsoft, Washington, DC, USA) and then imported into SPSS 21 (IBM, Armonk, NY, USA) using a general linear model (GLM) for the following model calculations. $Y_{ij} = \mu + \tau_i + \epsilon_{ij}$, where $Y_{ij}$ is the dependent variable, $\mu$ is the common effect of the whole experiment, $\tau_i$ represents the $i_{th}$ dietary effect, and $\epsilon_{ij}$ represents the random error present in the $j_{th}$ observation point of the $i_{th}$ diet. Gas production kinetics were calculated by the nonlinear procedure of the SAS Studio (Cary, NC, USA). A significant difference was declared at $p \leq 0.05$, and there is a significant difference between different groups marked with different lowercase letters within the same row in tables. Trends were recognized at $0.05 < p \leq 0.10$.

**3. Results**

*3.1. In Vitro Rumen Fermentation*

Table 4 and Figure 1 showed different silages for in vitro degradability and kinetic gas production. After 24 h of in vitro fermentation, IVDMD in the PVS1 and CS was significantly higher than PVS2 ($p < 0.01$). IVNDFD and IVADFD in the PVS1 were significantly higher than the other groups ($p < 0.01$). After 48 h of in vitro fermentation, IVDMD, IVNDFD, and IVADFD in the CS were significantly higher than the other groups ($p < 0.01$). In addition, gas production in the CS was significantly higher than PVS2 ($p < 0.01$). Further, the model established by Wolin $CH_4$ yield [22] in the silage fermentation process was predicted, and the $CH_4$ production in the CS group was significantly higher than in the PVS groups ($p < 0.001$). Regarding gas production kinetics, the highest theoretical maximum gas production, the maximum gas production rate, the maximum substrate digestion rate, and AGPR were significantly higher compared to CS ($p < 0.05$).

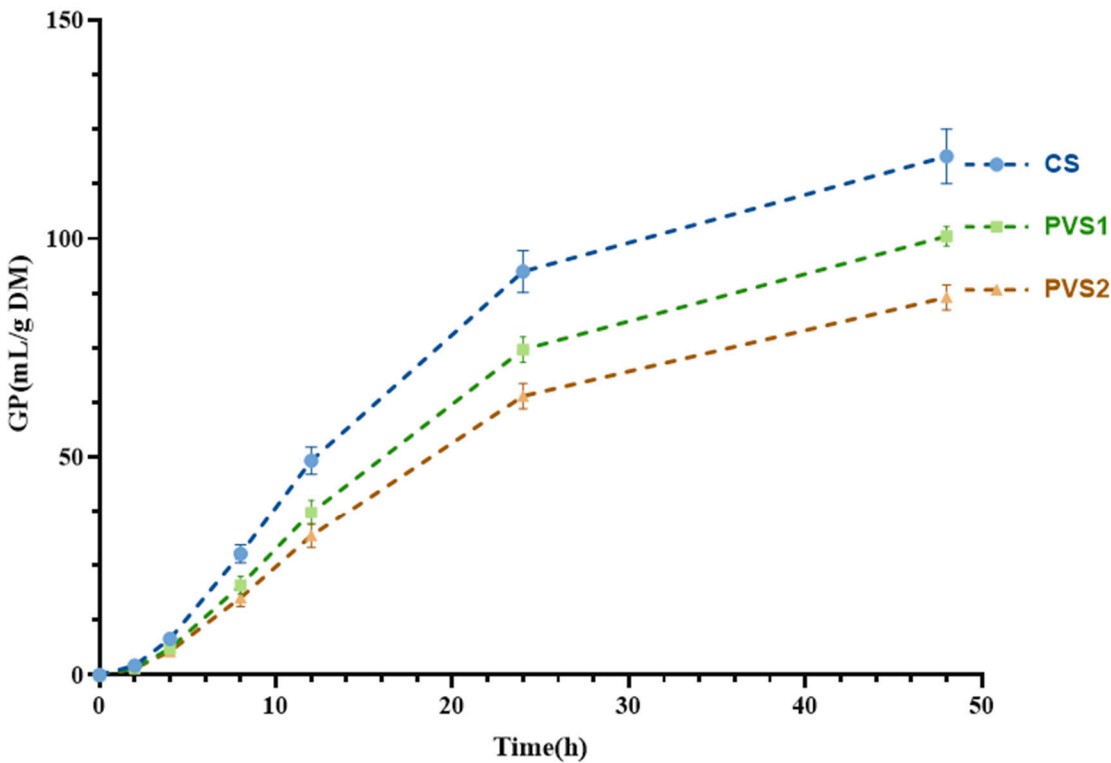

**Figure 1.** Gas emission profiles from the in vitro digestion of three different silage for 48 h.

**Table 4.** In vitro ruminal disappearance and gas production kinetics for three different silage.

| Items [1] | CS | PVS1 | PVS2 | SEM | *p*-Value [2] |
|---|---|---|---|---|---|
| Ruminal disappearance | | | | | |
| $IVDMD_{24}$/% | 66.66 [a] | 68.49 [a] | 64.13 [b] | 0.524 | 0.001 |
| $IVDMD_{48}$/% | 77.19 [a] | 73.74 [b] | 71.68 [c] | 0.519 | <0.001 |
| $IVNDFD_{24}$/% | 37.86 [b] | 41.80 [a] | 34.47 [b] | 0.923 | 0.002 |
| $IVNDFD_{48}$/% | 57.37 [a] | 50.62 [b] | 45.51 [c] | 1.088 | <0.001 |
| $IVADFD_{24}$/% | 31.40 [b] | 41.99 [a] | 29.82 [b] | 1.338 | <0.001 |
| $IVADFD_{48}$/% | 52.45 [a] | 49.79 [b] | 38.94 [c] | 1.290 | <0.001 |
| $GP_{48}$/(mL/g DM) [3] | 118.81 [a] | 100.49 [ab] | 88.70 [b] | 3.487 | 0.001 |
| Predicted $CH_4$ (48 h) (mL/g DM) | 21.42 [a] | 19.00 [b] | 14.99 [c] | 0.666 | <0.001 |
| Gas production kinetics | | | | | |
| A/(mL/g DM) | 131.40 [a] | 115.24 [ab] | 100.77 [b] | 3.865 | 0.005 |
| B | 2.01 | 1.97 | 2.03 | 0.021 | 0.530 |
| C/(h) | 15.56 | 17.44 | 17.59 | 0.407 | 0.078 |
| TRmaxG/(h) | 8.96 | 10.10 | 10.04 | 0.356 | 0.410 |
| RmaxG/(mL/h) | 5.51 [a] | 4.32 [b] | 3.67 [b] | 0.204 | <0.001 |

**Table 4.** *Cont.*

| Items [1] | CS | PVS1 | PVS2 | SEM | *p*-Value [2] |
|---|---|---|---|---|---|
| TRmaxS/(h) | 15.19 | 17.47 | 17.37 | 0.574 | 0.195 |
| RmaxS/(h) | 0.065 [a] | 0.058 [b] | 0.057 [b] | 0.001 | 0.012 |
| AGPR /(mL/h) | 4.24 [a] | 3.32 [b] | 2.86 [b] | 0.149 | 0.001 |

[1] A: the asymptotic gas production (mL/g DM); B: point of inflexion on curve parameter; C: the time (h) at which half of A is reached. TRmaxG: the time at which RmaxG is reached; RmaxG: the maximum gas production rate; TRmaxS: the time at which maximum rate of substrate digestion is reached; RmaxS: maximum substrate digestion rate; AGPR: the average gas production rate at the time when half of A occurred. [2] In the same row, values with no letter or the same letter superscripts mean no significant difference ($p > 0.05$), while with a different small letter superscripts mean significant difference ($p \leq 0.05$). The same as below. [3] The cumulative gas production data were calculated, the equation used was as below [21].

Table 5 shows the in vitro fermentation parameter of silage incubated with rumen fluids collected from Angus. After 24 h and 48 h of fermentation, pH in the CS was the lowest, which in the PVS2 was the highest ($p < 0.01$). $NH_3$-N in the PVS2 of culture fluids was significantly higher than PVS1 ($p < 0.05$). After 48 h of MCP in the PVS2, it was significantly higher than the CS ($p < 0.05$). For the VFA indexes, the total VFA in the CS was the highest, and propionate, butyrate, isovalerate, valerate, and TBCVFA were significantly higher than other groups ($p < 0.05$); acetate to propionate and NGR were significantly lower than other groups ($p < 0.01$) after 24 h and 48 h fermentation.

**Table 5.** In vitro rumen fermentation parameter of three different silage.

| Items [1] | CS | PVS1 | PVS2 | SEM | *p*-Value |
|---|---|---|---|---|---|
| Ruminal fermentation profile | | | | | |
| $pH_{24}$ | 6.27 [c] | 6.40 [b] | 6.54 [a] | 0.024 | <0.001 |
| $pH_{48}$ | 6.19 [c] | 6.42 [b] | 6.54 [a] | 0.030 | <0.001 |
| $NH_3$-$N_{24}$/(mg/dL) | 6.55 [ab] | 5.88 [b] | 7.73 [a] | 0.240 | 0.007 |
| $NH_3$-$N_{48}$/(mg/dL) | 8.48 [ab] | 8.04 [b] | 10.57 [a] | 0.421 | 0.020 |
| $MCP_{24}$/(μg/mL) | 331.31 | 334.33 | 345.83 | 3.385 | 0.173 |
| $MCP_{48}$/(μg/mL) | 308.71 [b] | 318.71 [ab] | 324.00 [a] | 2.564 | 0.040 |
| VFA pattern/(mmol/L) | | | | | |
| $Acetate_{24}$ | 19.38 [a] | 16.62 [ab] | 13.41 [b] | 0.831 | 0.007 |
| $Acetate_{48}$ | 25.46 [a] | 22.57 [a] | 18.96 [b] | 0.785 | 0.001 |
| $Propionate_{24}$ | 9.92 [a] | 7.39 [b] | 6.03 [b] | 0.495 | 0.002 |
| $Propionate_{48}$ | 12.20 [a] | 9.61 [b] | 8.43 [b] | 0.417 | <0.001 |
| $Isobutyrate_{24}$ | 0.15 | 0.15 | 0.10 | 0.015 | 0.341 |
| $Isobutyrate_{48}$ | 0.27 | 0.22 | 0.25 | 0.010 | 0.198 |
| $Butyrate_{24}$ | 2.40 [a] | 1.89 [b] | 1.80 [b] | 0.106 | 0.036 |
| $Butyrate_{48}$ | 3.45 [a] | 2.69 [b] | 2.55 [b] | 0.116 | 0.001 |
| $Isovalerate_{24}$ | 0.45 [a] | 0.28 [b] | 0.28 [b] | 0.024 | 0.001 |
| $Isovalerate_{48}$ | 0.65 [a] | 0.49 [b] | 0.53 [b] | 0.023 | 0.004 |
| $Valerate_{24}$ | 0.29 [a] | 0.20 [b] | 0.17 [b] | 0.016 | 0.003 |
| $Valerate_{48}$ | 0.41 [a] | 0.29 [b] | 0.28 [b] | 0.016 | <0.001 |

**Table 5.** *Cont.*

| Items [1] | CS | PVS1 | PVS2 | SEM | *p*-Value |
|---|---|---|---|---|---|
| Total VFA$_{24}$ | 32.60 [a] | 26.53 [b] | 21.79 [b] | 1.458 | 0.005 |
| Total VFA$_{48}$ | 42.43 [a] | 35.89 [b] | 30.99 [c] | 1.331 | <0.001 |
| Acetate to propionate$_{24}$ | 1.97 [b] | 2.28 [a] | 2.23 [a] | 0.044 | 0.003 |
| Acetate to propionate$_{48}$ | 2.09 [b] | 2.34 [a] | 2.25 [a] | 0.029 | <0.001 |
| TBCVFA$_{24}$ | 0.60 [a] | 0.43 [b] | 0.38 [b] | 0.033 | 0.009 |
| TBCVFA$_{48}$ | 0.92 [a] | 0.72 [b] | 0.78 [b] | 0.031 | 0.012 |
| NGR$_{24}$ | 2.41 [b] | 2.75 [a] | 2.77 [a] | 0.051 | 0.001 |
| NGR$_{48}$ | 2.61 [b] | 2.85 [a] | 2.80 [a] | 0.033 | 0.011 |

[1] TBCVFA: total branched volatile fatty acid; NGR: the ratio of non-glucogenic to glucogenic acid, the equation used was as below [22]: NGR = (Acetate + 2 × Butyrate + Valerate)/(Propionate + Valerate).

### 3.2. In Vivo Rumen Fermentation

The effect of adding different silage types to the diet on rumen fermentation parameters is shown in Table 6. Neither the addition of PVS1 nor PVS2 to the diet affected VFA in the rumen fluid, the rumen pH of the beef cattle, or the concentration of ammoniacal nitrogen in the rumen fluid. However, there was a tendency to increase the acetate to propionate ($p < 0.1$).

**Table 6.** Rumen fermentation parameter in three different dietary silage groups.

| Items | CS | PVS1 | PVS2 | SEM | *p*-Value |
|---|---|---|---|---|---|
| pH | 6.49 | 6.60 | 6.64 | 0.07 | 0.652 |
| NH$_3$-N mg/dL | 7.03 | 6.42 | 6.37 | 0.21 | 0.391 |
| VFA pattern (mmol/L) | | | | | |
| Acetate | 31.20 | 27.59 | 30.26 | 3.08 | 0.897 |
| Propionate | 8.71 | 6.83 | 7.49 | 0.88 | 0.704 |
| Isobutyrate | 0.28 | 0.31 | 0.32 | 0.28 | 0.830 |
| Butyrate | 4.92 | 4.20 | 4.55 | 0.49 | 0.852 |
| Isovalerate | 0.74 | 0.57 | 0.57 | 0.07 | 0.499 |
| Valerate | 0.52 | 0.43 | 0.46 | 0.04 | 0.583 |
| Total VFA | 46.38 | 39.94 | 43.65 | 4.54 | 0.860 |
| Acetate to propionate | 3.66 | 4.02 | 4.27 | 0.11 | 0.076 |
| TBCVFA | 1.02 | 0.88 | 0.89 | 0.09 | 0.788 |
| NGR | 4.55 | 5.02 | 5.27 | 0.13 | 0.062 |

### 3.3. Rumen Microbial Diversities

In the α diversity index, there were no significant differences among the microbial diversity indexes of ACE, chao1, Shannon, and PD_whole_tree in the rumen (Figure 2 and Supplementary Table S2). PCoA analysis showed that the CS group was significantly different from PVS diary groups in the β diversity analysis of the rumen microbial community (Figure 3).

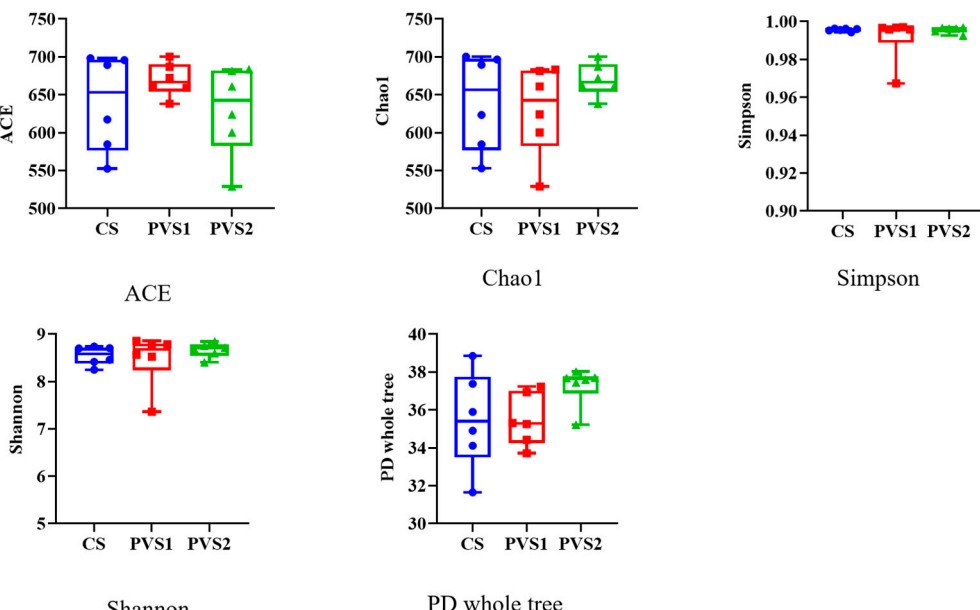

**Figure 2.** The α diversity index of rumen bacterial communities in three different dietary silage groups.

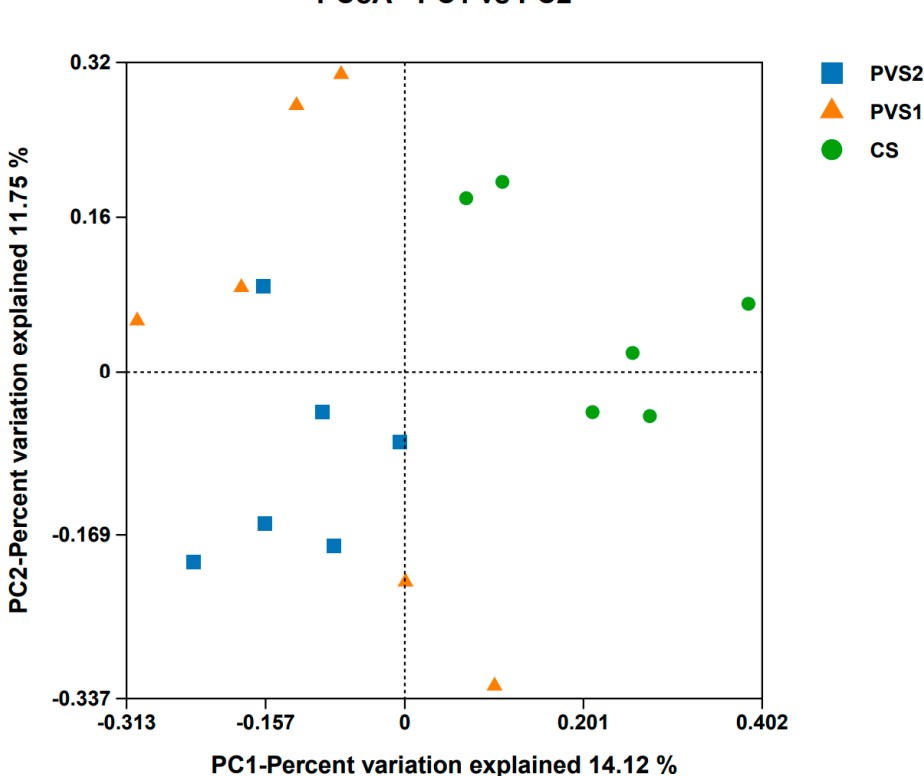

**Figure 3.** PCoA analysis of the rumen bacterial community structures and relationship in three different dietary silage groups.

### 3.4. Compositions of Rumen Microbiota

The analysis of bacterial classification at the phylum level (Figure 4a and Supplementary Table S3) showed 13 bacterial phyla with a relative abundance greater than 0.1%. Compared with the CS group, the abundance of proteobacteria in the rumen was significantly reduced in the PVS1 diet group ($p = 0.034$). The PVS group tended to reduce

Spirochaetota abundance ($p$ = 0.057) and increase Synergistota abundance ($p$ = 0.077) at the genus level (Figure 4b and Supplementary Table S4). Compared with the CS diet group, the two PVS groups significantly reduced the abundance of *Prevotella* ($p$ = 0.001) and tended to increase the abundance of *unclassified_F082* ($p$ = 0.064); the PVS1 diet group significantly increased the abundance of *uncultured_rumen_bacterium* ($p$ = 0.037); the PVS2 diet group significantly increased the abundance of *Lachnospiraceae_XPB1014_group* ($p$ = 0.040) and *Bacteroidales_bacterium_Bact_22* ($p$ = 0.013), but significantly decreased the abundance of *unclassified_Prevotellaceae* ($p$ = 0.007) in the rumen.

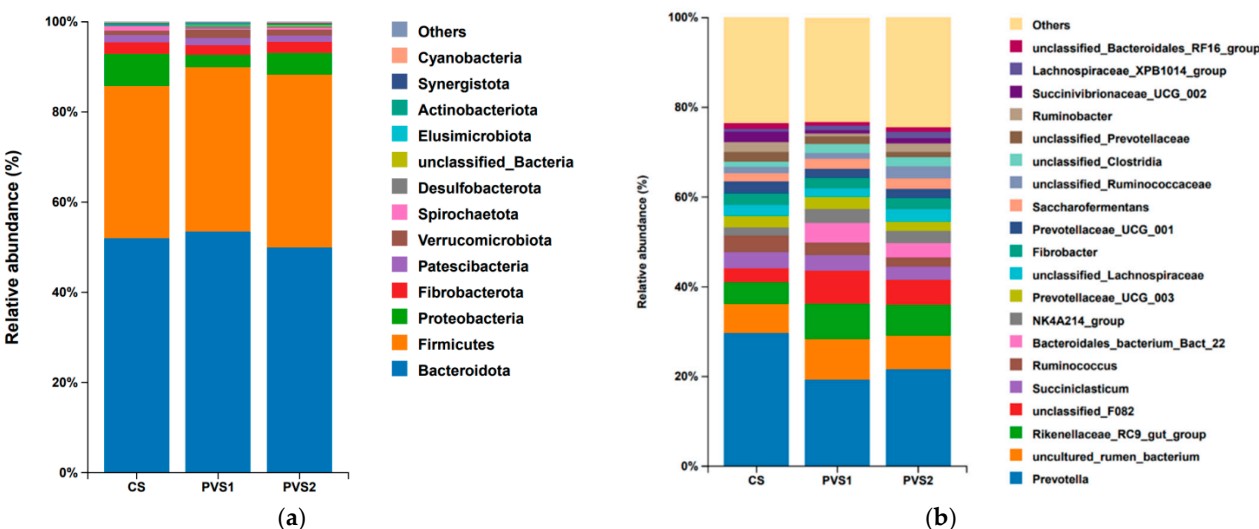

**Figure 4.** Microbial composition of three different dietary silage groups. (**a**) Taxa assignments at the phylum level (Relative abundance ≥ 0.1%). (**b**) Taxa assignments at the genus level (Relative abundance ≥ 1%). Each bar represents the average relative abundance of each bacterial taxon within a group.

Through LEfSe analysis, it was found that under the experimental conditions, one phylum, one class, one order, four families, two genuses, and two species caused the main differences in rumen microorganisms (Figure 5). At the phylum level, proteobacteria was enriched in the CS group; at the class level, *Gammaproteobacteria* was enriched in the CS group. At the order level, *Enterobacterales* were enriched in the CS group; at the family level, *Prevotellaceae* and *Succinivibrionaceae* were enriched in the CS group, *F082* was enriched in the PVS1 group, and *Oscillospiraceae* was enriched in the PVS2 group At the genus level, *Prevotella* was enriched in the CS group, and *Bacteroidales_bacterium_Bact_22* was enriched in the PVS1 group; at the species level, *unclassified Prevotella* was enriched in the CS group, *unclassified Bacteroidales_bacterium_Bact_22* was enriched in the PVS1 group.

### 3.5. Correlation Analysis

The relationships among growth performance, rumen fermentation indexes, and bacteria with abundance ratios higher than 1% at the genus level were evaluated (Figure 6). The results showed that $NH_3$-N was negatively correlated with the abundance of *unclassified_F082* (r = −0.53; $p$ < 0.05). The abundance of *Rikenellaceae_RC9 gut_group* was negatively correlated with acetate (r = −0.69; $p$ < 0.01), propionate (r = −0.72; $p$ < 0.001), butyrate (r = −0.57; $p$ < 0.05), valerate (r = −0.71; $p$ < 0.01), isovalerate (r = −0.72; $p$ < 0.001), TVFA (r = −0.69; $p$ < 0.01), TBCVFA (r = −0.66; $p$ < 0.01), and positively correlated with acetate to propionate (r = 0.63; $p$ < 0.01) and NGR (r = 0.71; $p$ < 0.01) in the rumen. The abundance of *Prevotella* was positively correlated with acetate (r = 0.51; $p$ < 0.05), propionate (r = 0.56; $p$ < 0.05), butyrate (r = 0.60; $p$ < 0.01), valerate (r = 0.55; $p$ < 0.05), isovalerate (r = 0.49; $p$ < 0.05) and TVFA (r = 0.54; $p$ < 0.05) in the rumen. The abundance of *Ruminococcus* was

negatively correlated with acetate to propionate (r = −0.49; $p < 0.05$) and NGR (r = −0.55; $p < 0.05$) in the rumen fluid.

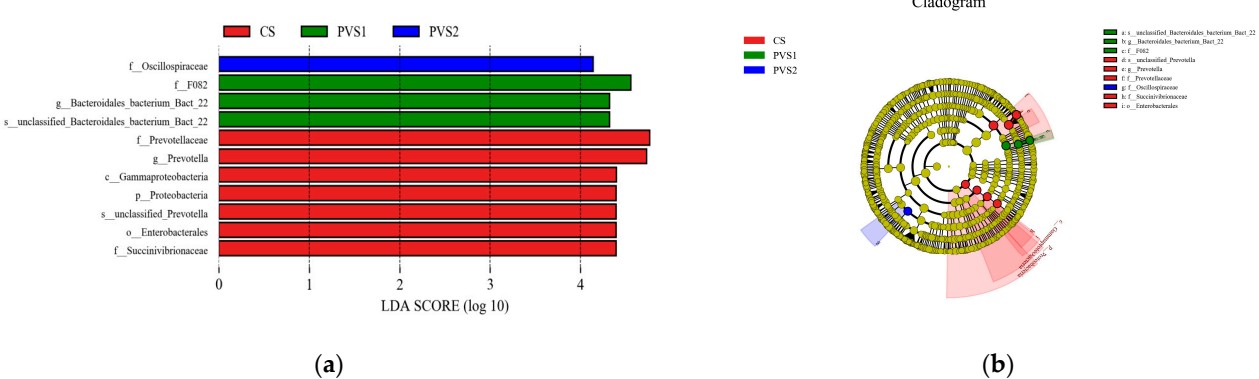

(**a**)　　　　　　　　　　　　　　　　　　(**b**)

**Figure 5.** Linear discriminant analysis effect size (LEfSe) of the rumen bacterial taxa in three different dietary silage groups. (**a**) Histogram of the LDA scores computed or bacterial taxa differentially abundant among different dietary groups (LDA threshold 4). (**b**) LEfSe analysis clade diagram. Cluster tree for LEfSe analysis, different colors represent different groupings, and nodes of different colors represent the microbiome that plays an important role in the grouping represented by the color.

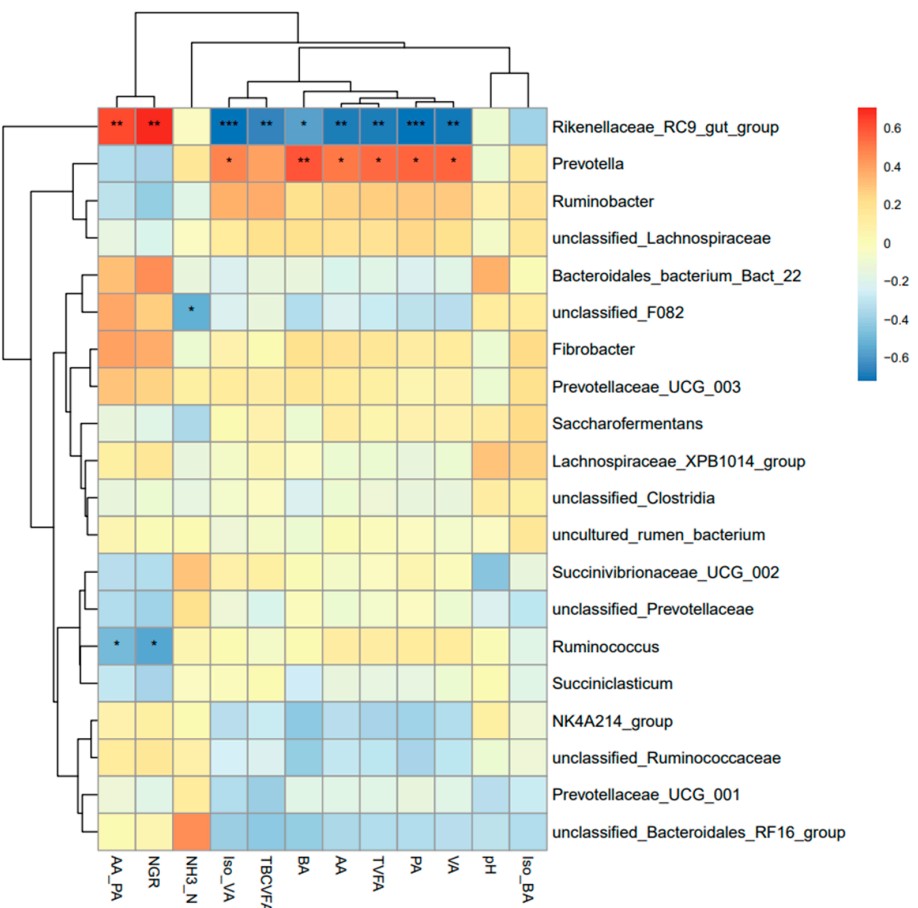

**Figure 6.** Correlation of rumen fermentation indexes and the rumen flora of different dietary groups. Strong correlations are shown in red and blue, where 1 is a completely positive correlation (dark red), −1 is a negative correlation (dark blue), and weak correlations are shown in yellow. Pearson test: *, $p \le 0.05$; **, $p \le 0.01$; ***, $p \le 0.001$. AA: acetate; PA: propionate; BA: butyrate; Iso_BA: isobutyrate; VA: valerate; Iso_VA: isovalerate; AA_PA: acetate to propionate; TVFA: total volatile fatty acids; TBCVFA: total branched-chain volatile fatty acids.

### 3.6. Metabolic Functions

The PICRUSt2 functional prediction (KEGG) results are shown in Table 7. The PVS group had significantly higher abundances in the pathways related to microbial metabolism in diverse environments, including carbon metabolism, ribosome, aminoacyl-tRNA biosynthesis, carbon fixation pathways in prokaryotes, peptidoglycan biosynthesis, protein export, RNA degradation, fatty acid biosynthesis, fatty acid metabolism, and terpenoid backbone biosynthesis ($p < 0.05$). The abundance of pathways in the CS group was related to amino sugar and nucleotide sugar metabolism, and lysine biosynthesis was significantly higher than in the PVS1 group, but the abundance of glycine, serine, and threonine metabolism was the opposite. Additionally, except for the abundance of pathways mentioned above, the highest was in the CS group ($p < 0.05$).

**Table 7.** The relative abundance of KEGG pathways of bacteria in rumen of different dietary groups.

| Items | CS | PVS1 | PVS2 | SEM | *p*-Value |
|---|---|---|---|---|---|
| Metabolic pathways | 17.62 [a] | 17.49 [b] | 17.45 [b] | 0.028 | 0.021 |
| Biosynthesis of secondary metabolites | 8.01 [a] | 7.93 [b] | 7.95 [b] | 0.014 | 0.006 |
| Microbial metabolism in diverse environments | 3.69 [b] | 3.76 [a] | 3.74 [a] | 0.009 | <0.001 |
| Carbon metabolism | 2.57 [b] | 2.64 [a] | 2.62 [a] | 0.010 | 0.002 |
| Ribosome | 2.55 [b] | 2.62 [a] | 2.60 [a] | 0.010 | 0.007 |
| Amino sugar and nucleotide sugar metabolism | 1.24 [a] | 1.19 [b] | 1.21 [ab] | 0.008 | 0.046 |
| Aminoacyl-tRNA biosynthesis | 1.08 [b] | 1.12 [a] | 1.12 [a] | 0.005 | 0.001 |
| Alanine, aspartate and glutamate metabolism | 1.00 [a] | 0.97 [b] | 0.97 [b] | 0.004 | 0.002 |
| Carbon fixation pathways in prokaryotes | 0.94 [b] | 0.98 [a] | 0.96 [a] | 0.006 | 0.003 |
| Starch and sucrose metabolism | 0.99 [a] | 0.89 [b] | 0.91 [b] | 0.013 | 0.001 |
| Glycine, serine and threonine metabolism | 0.82 [b] | 0.84 [a] | 0.83 [ab] | 0.003 | 0.043 |
| Peptidoglycan biosynthesis | 0.75 [b] | 0.76 [a] | 0.76 [a] | 0.001 | 0.041 |
| Fructose and mannose metabolism | 0.77 [a] | 0.73 [b] | 0.73 [b] | 0.008 | 0.018 |
| Galactose metabolism | 0.72 [a] | 0.65 [b] | 0.67 [b] | 0.010 | 0.002 |
| Protein export | 0.60 [b] | 0.62 [a] | 0.61 [a] | 0.002 | 0.005 |
| RNA degradation | 0.59 [b] | 0.61 [a] | 0.60 [a] | 0.004 | 0.031 |
| Lysine biosynthesis | 0.60 [a] | 0.58 [b] | 0.60 [ab] | 0.003 | 0.041 |
| Pantothenate and CoA biosynthesis | 0.60 [a] | 0.59 [b] | 0.58 [b] | 0.002 | 0.009 |
| Fatty acid biosynthesis | 0.57 [b] | 0.60 [a] | 0.59 [a] | 0.006 | 0.025 |
| Fatty acid metabolism | 0.55 [b] | 0.60 [a] | 0.59 [a] | 0.006 | 0.003 |
| Terpenoid backbone biosynthesis | 0.55 [b] | 0.56 [a] | 0.56 [a] | 0.002 | 0.001 |
| Nicotinate and nicotinamide metabolism | 0.55 [a] | 0.54 [b] | 0.54 [b] | 0.002 | 0.015 |

Only the KEGG pathways with a relative abundance above 0.5% and significant difference ($p \leq 0.05$) is presented.

## 4. Discussion

### 4.1. In Vitro Rumen Fermentation Characteristic of Different Silages

In this study, 24 h and 48 h of IVDMD, IVNDFD, and IVADFD in the CS group were significantly higher than in the PVS groups. This may be because potato vine and leaf are harvested close to the ground, resulting in higher ash content and lower starch content, which reduce the disappearance of DM, NDF, and ADF in the rumen [24]. During the fermentation process of the diet, rumen microorganisms consume carbohydrates and other nutrients that are ingested to produce $CH_4$, $H_2$, $CO_2$ and other gases. The gas production within a certain period reflects the utilization degree of the substrate by rumen microorganisms and the substrate [25]. Compared with the CS group, the gas production, predicted $CH_4$ production, and AGPR of the PVS groups were significantly decreased compared to the CS group, which was similar to the results of Guo [10]. This reduction in gas production could be due to decreased DM digestibility, as the total starch content and DM digestibility of rumen fermentation substrates significantly affect fermentation gas production [26]. Moreover, phenolic compounds and protease inhibitors contained in the potato vine and leaf may also affect gas production kinetics [27]. Although silage potato

vine has been found to significantly reduce its glycoalkaloid content [4,28], the remaining glycoalkaloids, and other secondary metabolites may affect the activity of some rumen microorganisms, resulting in low in vitro gas production [24].

After 48 h of the rumen in vitro fermentation, the pH value of each group exceeded 6, indicating a stable and healthy in vitro rumen fermentation environment [29]. VFAs are an intermediate metabolite of rumen energy utilization in ruminants and are an important indicator reflecting the digestion and metabolism of the rumen, which can provide 70% to 80% of the total energy for ruminants. The yield and component of VFA can reflect the metabolic status of rumen microorganisms [30]. The pH value of the CS group was significantly lower than the PVS group because the CS group had a higher starch content which produced more VFA and lactic acid during in vitro fermentation, resulting in a decrease in the pH value. However, the acetate to propionate in the CS group was significantly lower than that in the PVS group, and propionic acid was significantly higher than in the PVS group. This was due to a higher starch content in the CS group [31]. However, the AGPR of the CS group was significantly higher than the PVS1 group, and there was no significant difference in acetate between the two groups. Branched-chain VFA can stimulate the activity of crude fiber-decomposing bacteria, increase the biomass of structural carbohydrate-decomposing bacteria, improve DM digestibility, and increase gas production [32,33]. Isovalerate and TBCVFA in the CS group were significantly higher than in the PVS group, which was the reason for a reduction in AGPR in the PVS group.

$NH_3$-N in the rumen is an important product of the nitrogen metabolism process in the rumen, and it serves as a source of nitrogen for the growth of rumen microorganisms [34]. Thus, the $NH_3$-N concentration in rumen fluid reflects the balance between protein degradation and synthesis in the rumen. After 48 h of fermentation, the MCP of PVS2 was significantly higher than CS, and $NH_3$-N was significantly higher than PVS1. However, there was no significant difference between the MCP of PVS1 and PVS2. The MCP of PVS2 was significantly higher than CS, and the $NH_3$-N was significantly higher than PVS1. The MCP of the PVS1 and PVS2 was not significantly higher than the CS group, which could be related to the addition of the potato vine and leaf containing more CP than CS in the diet. However, the inclusion of potato vine and leaf into the silage resulted in a reduction in both the starch content and ME of the silage. However, the increased CP content in PVS could enhance the balance of the energy-nitrogen ratio within the fermentation substrate [35]. The increased CP can also promote the growth of rumen microorganisms, accelerate the transformation of nitrogen-containing substances in the substrate and the synthesis of MCP, and further improve the nitrogen fixation efficiency of microorganisms [36–38].

### 4.2. Rumen Fermentation Parameter in Different Dietary Groups

Beef cattle's normal rumen pH value is between 5.8 and 6.8 [15], which can maintain the normal activities and growth of rumen microorganisms. Under the conditions of this experiment, the pH value of Angus rumen fluid ranged from 6.49 to 6.64. However, there were no significant differences between the CS and PVS group regarding rumen pH, $NH_3$-N, and the composition of rumen VFA; however, there was a tendency to increase the acetate to propionate. This result is consistent with the experimental results of Liang [39], who fed lambs with different proportions of PVS instead of corn silage. The increased acetate to propionate may be due to the low starch content in the two groups fed PVS, which is consistent with the results of the in vitro gas production test of the silage raw material.

### 4.3. Rumen Bacteria Composition in Different Dietary Groups

The rumen microbial community is a complex network that breaks down the various roughages ingested by fermented ruminants. The rumen community is closely related to the host's physiological structure and dietary components [40,41]. The dominant microorganisms are responsible for digesting large amounts of protein, cellulose, and starch in roughage, which are the substrates used for the main production capacity of microbial

growth [42]. The α diversity of microorganisms reflects the diversity within a community, which is mainly related to the number of species and diversity [43]. In this study, there was no significant difference in the α diversity of rumen microorganisms between groups, indicating that the three diets did not affect the number and diversity of rumen microorganisms. Microbial β diversity reflects whether there are significant differences in microbial communities between multiple samples and measures the degree of change in sample diversity under different factors. In this study, the PCoA results revealed significant separation between the CS and PVS groups, with a significantly different rumen microorganism community composition, suggesting that feeding silage made of the potato vine and leaf changed the rumen microbial community composition of beef cattle. However, the two groups of PVS did not show a separate state, and the community compositions were similar. Furthermore, LEfSe analysis found significant differences in the abundance of many species between groups.

In this study, the most abundant flora was Bacteroidetes, Firmicutes, and Proteobacteria: the bacterial phyla that play an important role in rumen fermentation [44–46]. Among them, the abundance of Proteobacteria was significantly increased in the CS group with a lower CP content. This result was consistent with Luo's results, which found that the abundance of Proteobacteria decreased with an increasing CP content in the diet [43].

Bacteroides can degrade cellulose in feed because their genome can express enzymes that degrade plant polysaccharides [47,48]. In this study, the relative abundance of *Prevotella* and *unclassified_Prevotellaceae* in the PVS group was significantly lower compared to the CS group. In ruminants, *Prevotella* has been reported to have a crucial role in modulating the activity of type IV dipeptidase, which is primarily involved in the cleavage of oligopeptides [49]. This suggests that *Prevotella* is instrumental in the efficient breakdown of oligopeptides and plays a significant role in rumen protein metabolism and $NH_3$-N concentration [49,50]. The decreased relative abundance of *Prevotella* may explain the decrease in $NH_3$-N concentrations. Additionally, the correlation analysis revealed a significantly positive correlation between *Prevotella* and the rumen fermentation index VFA content ($r^2 > 0.4$; $p < 0.05$) because *Prevotella* is closely related to the digestion of carbohydrates and fiber [51–53]. The higher relative abundance of *Prevotella* in the CS group promoted the digestion and utilization of carbohydrates, protein, and fiber in the rumen and facilitated the synthesis of VFA. This is consistent with the results of PICRUSt2, which showed that CS group diets increased the relative abundance of carbohydrate metabolic pathways. The relative abundance of the alanine, aspartate, and glutamate metabolism pathways was higher in the CS group, possibly due to the higher alanine, aspartate, and glutamate contents in the amino acid composition of whole-plant corn silage [54]. In this trial, the relative abundance of *Bacteroidales_bacterium_Bact_22* in the PVS group was significantly increased. It was positively correlated with the rumen papillae length, which could potentially enhance nutrient absorption through the rumen wall [55,56]. Additionally, PICRUSt2 functional prediction (KEGG) further confirmed the upregulation of fatty acid biosynthesis and metabolism pathways in the PVS groups. However, there was no significant difference in the content of VFA in the rumen, which could be due to microbial synthesis since VFA can be rapidly used to provide metabolic energy to the host through simple diffusion or vector-mediated transport [57,58]. Similarly, the relative abundance of *F082* in PVS showed an increasing trend compared to the CS group. A study found that *F082* in the gut plays an important role in protein metabolism [59], but its exact role in the rumen is unclear. Interestingly, in this trial, the CP content of the PVS was higher than CS, but a significant negative correlation was observed between CP and $NH_3$-N in the rumen. Combined with the results of the in vitro fermentation of $NH_3$-N, it was suggested that bacteria may be involved in the metabolism of protein and nitrogen-containing substance transformation in the rumen. In addition, the correlation analysis found that *Rikenellaceae_RC9_gut_group* was significantly and negatively correlated with the content of the rumen fermentation index VFA ($r^2 < -0.5$; $p < 0.05$) because it could degrade cellulose and hemicellulose and promote the synthesis of VFA [60–62].



Regarding Firmicutes, *Succinivibrionaceae* was enriched in the CS group, which could degrade nonstructural carbohydrates [41] and participate in the fermentation of succinate [63] and its conversion to propionate. In this trial, the acetate-to-propionate rumen fermentation of the CS group tended to increase compared to the other groups, reflecting this feature. Additionally, the relative abundance of *Lachnospiraceae_XPB1014_group* increased significantly in the PVS2 group. It was found that *Lachnospiraceae_XPB1014_group* played an important role in alkaloid degradation [64]. *Oscillospiraceae* was also concentrated in the PVS2 group, with the ability to degrade proteins, including the protein backbone of mucin [65]. Potatoes contain mucin-17 [Solanum tuberosum (potato)] [66]. The increase in the abundance of the above two bacteria may be to better degrade the toxic substances remaining in the potato silage process. Correlation analysis found that *Ruminococcus* abundance was inversely correlated with acetic acid/propionate and NGR ($r^2 < -0.4$; $p < 0.05$). This was because some species in *Ruminococcus*, such as *Ruminococcus flavefaciens*, synthesize succinate as the final product [67], which provides *Succinivibrionaceae* with a substrate for the conversion of propionic acid, promoting the production of propionate.

Spirochaetota is involved in pectin degradation, which is particularly abundant in diets that are high in pectin [68,69]. The increased Spirochaetota abundance in the rumen of the CS group may have facilitated the more effective degradation of dietary fiber in the CS group. Synergistota, first found in goat rumen, is often associated with the degradation of toxic compounds [70]. The increase in the relative abundance of Synergistota and *Lachnospiraceae_XPB1014_group* in the PVS group illustrated that although PVS was found to reduce its glycoalkaloid content significantly, the remaining glycoalkaloids and other secondary metabolites could affect the activity of some rumen microorganisms, and the degradation of the above bacteria was still required to maintain a normal rumen fermentation environment. Synergistota has a limited ability to degrade polysaccharides because recent studies have found that its abundance increases with an increase in cellulose digestibility [71,72], suggesting that it may play an auxiliary or complementary role in carbohydrate degradation and metabolism in the rumen [73]. These observations are further supported by the PICRUSt2 functional prediction (KEGG) results, which in PVS are relatively abundant with microbial metabolic pathways.

## 5. Conclusions

Under the conditions of this trial, the in vitro gas production fermentation result showed that the potato vine and leaf mixed silage could significantly reduce gas and CH4 production. Compared with the traditional whole-plant corn silage diet, feeding potato vine and leaf mixed silage had no negative effect on rumen fermentation or the rumen microbial community. Combining the results of rumen fermentation in vitro and in vivo with potato vine and leaf mixed silage production provides new and sustainable insight into potato vine and leaf utilization.

**Supplementary Materials:** The following supporting information can be downloaded at: https://www.mdpi.com/article/10.3390/fermentation9080704/s1, Table S1: Buffer solution composition; Table S2: The diversity index of bacterial communities in the rumen in different dietary groups; Table S3: The relative abundance of dominant bacteria was ≥0.1% (phylum level); Table S4: The relative abundance of dominant bacteria was ≥1% (family level).

**Author Contributions:** Conceptualization, S.Z., J.D., B.C. and H.S.; Methodology, S.Z., J.D., Y.H., B.C. and H.S.; Software, S.Z. and J.D.; Validation, S.Z., Y.C. and Y.H.; Formal analysis, S.Z. and J.D.; Investigation, S.Z., J.D., Y.C., L.W., Y.L. (Yingqi Li), S.M., H.W., Q.Z., P.L. and Y.L. (Yawen Luo); Resources, S.Z., J.D., Y.L. (Yingqi Li), X.W. and H.S.; Data curation, S.Z. and J.D.; Writing—original draft, S.Z. and J.D.; Writing—review & editing, S.Z., J.D., X.Q., Y.H. and H.S.; Visualization, S.Z., J.D. and P.L.; Supervision, Y.H. and H.S.; Project administration, B.C. and H.S.; Funding acquisition, H.S. All authors have read and agreed to the published version of the manuscript.

**Funding:** This work was funded by the National Key R&D Program of China (2022YFD1602310 & 2022YFD1601308), the Key Technology R&D Program of Ningxia (2017BY078), and the China Agriculture Research Systems of MOF and MARA (CARS-37).

**Institutional Review Board Statement:** All the cattle in our experiment were compliant with the Guidelines of the Animal Care Com-mittee and animal welfare guidelines of China Agricultural University (AW82303202-1-1).

**Informed Consent Statement:** Not applicable.

**Data Availability Statement:** The datasets generated during and/or analyzed during the current study are available from the corresponding author on reasonable request.

**Acknowledgments:** Part of this work was assisted by the Fangshan Beef Cattle Experimental Base and Benwang Farm in China.

**Conflicts of Interest:** The authors declare that they have no conflict of interest.

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
