# Peer review of "Effect of Potato Vine and Leaf Mixed Silage to Whole Corn Crops on Rumen Fermentation and the Microbe of Fatten Angus Bulls"

_fermentation, doi:10.3390/fermentation9080704_

Round 1

Reviewer 1 Report

Dear Authors,

You performed an interesting study.

However, the introduction is too general. Please precise. Give some more details also in the methods part and include the supplementary tables 1-3 in the main text. Please explain what you mean exactly by  jujube powder, stone meal, bran, fermented feed and corn (which kind or plant parts etc.).

The starch content of the CS diet is extremely high and beyond recommended limits!

Kind regards,

The reviewer

Please look for someone to edit the language properly.

Author Response

As attached FYI~

Reviewer 2 Report

The manuscript (fermentation-2484830) describes a trial aimed to investigate how the use of potato vine and leaf mixed silage affects the rumen fermentation characteristics, both in vitro and in vivo. The findings of this study can serve as a guide for assessing the effectiveness and worth of potato vine and leaf as a substitute for corn silage, as well as for mitigating greenhouse gas emissions. The manuscript presents a well-conceived idea. However, it requires extensive revision to improve the organization, clarity, and coherence of the text.

-   The title does not represent the experimental design and the objective of the study accurately. Please rewrite

-   Lines 11-16: This sentence may be more effective if it is broken down into several shorter sentences to improve readability and clarity.

-   Abstract: The results are concise and not representative of the study. Please add more information about the results obtained.

-   Lines 50-52: “As 2014 FAO research, agricultural activities account for about 24% of global greenhouse gas emissions (GHG), …… activities [12]” It may be beneficial to update the citation for this statement, as the 2014 FAO research may not reflect the most current information available. Perhaps a more recent source could be used to support this claim.

-  The hypothesis of the study should be clarified at the end of the Introduction section.

-  Line 88 “Three Angus gelding bulls with permanent ruminal fistulas were selected” Please add the overall mean ± standard error.

- Lines 88-89: “(Fangshan Beef Cattle Experimental Base of China Agricultural University)” add more information such as city and country.

-   Lines 90-91: “…….the eighth edition of Nutrient Requirements of Beef Cattle (2016).” add to the references list.

-   Line 101: “A total of 1 g of the substrate” Please clarify if the substrate was dry or fresh.

-   Line 102: What is the composition of the buffer solution?

-  Lines 110-113: the sampling of MCP, NH3-N, and VFA is missing. Also, how many bottles were used in each parameter per time?

-   Line 115: description and the total number of animals should be clearly stated.

-   How were gas production kinetics calculated? Please clarify in the M&M section.

-   In all Tables and Figs., describe the experimental groups and all abbreviations used in the table's footnotes and figure legends.

-  Figure 1: the standard error bar is missing.

- Table 3: Acetate/propionate data is wrongly computed please revise your calculations.

-    Figure 2: What does the y-axis indicate?

-   In the conclusion section: The authors should be clearer and more direct. The conclusion should answer the aim of the study. Add also, the future perspective must be clarified.

-

Author Response

As attached FYI~

Round 2

Reviewer 2 Report

The authors adequately responded to all comments and performed all required modifications.